# Clinical Aspects of Gut Microbiota in Hepatocellular Carcinoma Management

**DOI:** 10.3390/pathogens10070782

**Published:** 2021-06-22

**Authors:** Jinghang Xu, Qiao Zhan, Yanan Fan, Emily Kwun Kwan Lo, Fangfei Zhang, Yanyan Yu, Hani El-Nezami, Zheng Zeng

**Affiliations:** 1Department of Infectious Diseases, Peking University First Hospital, Peking University, Beijing 100034, China; ddcatjh@sina.com (J.X.); dralettazhan@bjmu.edu.cn (Q.Z.); franc0101@163.com (Y.F.); yyy@bjmu.edu.cn (Y.Y.); 2School of Biological Sciences, University of Hong Kong, Pokfulam 999077, Hong Kong, China; emilylkk@hku.hk (E.K.K.L.); u3567792@connect.hku.hk (F.Z.); 3Institute of Public Health and Clinical Nutrition, School of Medicine, University of Eastern Finland, FI-70211 Kuopio, Finland

**Keywords:** hepatocellular carcinoma, dysbiosis, microbiota, probiotics

## Abstract

Liver cancer, predominantly hepatocellular carcinoma (HCC), is the third leading cause of cancer-related deaths worldwide. Emerging data highlight the importance of gut homeostasis in the pathogenesis of HCC. Clinical and translational studies revealed the patterns of dysbiosis in HCC patients and their potential role for HCC diagnosis. Research on underlying mechanisms of dysbiosis in HCC development pointed out the direction for improving the treatment and prevention. Despite missing clinical studies, animal models showed that modulation of the gut microbiota by probiotics may become a new way to treat or prevent HCC development.

## 1. Introduction

Liver cancer, predominantly hepatocellular carcinoma (HCC), is a substantial health burden worldwide. In 2017, with an estimation of 803,400 (753,100 to 856,700) cases, the age-standardized years lived with disability (YLDs) rate increased by 8.1% when compared with that in 2007 [1]. With a new death of 830,180 cases in 2020, liver cancer represents the third (8.3%) leading cause of cancer-related deaths worldwide [2]. Due to the low screening rate in high-risk populations and inadequate sensitivity of the present diagnostic technology (imaging and serum alpha-fetoprotein [AFP] quantification), HCC is usually diagnosed at the late stages, leading to low accessibility of curative therapy and high mortality. Early diagnosis and better prevention and treatment are the goals pursued by doctors and patients together. In terms of diagnostic technology, sensitive and specific biomarkers for early diagnosis of HCC are still lacking. As for prevention and treatment of HCC, apart from etiological treatment of HCC, such as anti-hepatitis B virus (HBV) in HBV-HCC, extra measures are in great need.

Approximately 4 × 10^13^ microbial cells spanning ~3 × 10^3^ species inhabit the human body. The vast majority (97%) of them are bacteria in the colon, and the remaining include extracolonic bacteria and Archaea and eukaryotes such as fungi [3,4]. Gut and liver are closely related, not only anatomically but also functionally. The liver receives blood from the gut through the portal vein, while the gut receives bile from the liver through the bile duct. Blood from the gut brings nutrition, microbial metabolite, and microbe-associated molecular patterns (MAMPs). MAMPs may elicit inflammatory responses by activating pattern recognition receptors (PRRs) in the liver, contributing to the progression of liver diseases and development of HCC. Bile acids, important components in bile, are synthesized from cholesterol in the liver, then metabolized by gut bacteria. They can shape the composition and function of the intestinal microbiota. Mutual interplay of bile acids and gut microbiota regulates many physiological processes [5,6]. Emerging data highlight the importance of gut homeostasis in the pathogenesis of HCC. Clinical and translational studies revealed the patterns of dysbiosis in HCC patients, indicating the diagnostic value of the dysbiosis in early diagnosis of HCC. Mechanism research demonstrates that gut microbiota plays an important role in liver tumorigenesis, which suggests the possibility of preventing and treating HCC by modulating gut microbiota.

Although the relationship between gut bacterial microbiota and fibrosis/liver cirrhosis is of importance to understand between gut bacterial microbiota and HCC, previous reviews have discussed this topic in detail [7,8]. Therefore, in the present review, we only focus on the alteration of gut bacterial microbiota in HCC patients and the underlying mechanisms of dysbiosis in HCC development. Meanwhile, diagnostic value of gut dysbiosis and therapeutic potential by targeting gut dysbiosis in HCC were discussed.

## 2. Gut Microbiota Changes in HCC Patients

Gut bacteria dysbiosis in HCC patients has been reported in many countries and regions recently (Table 1). Both stool and blood samples possess the value of diagnosing and assessing dysbiosis in HCC patients.

In the early stage, the number of colony-forming units per gram (cfu/g) of wet feces was adopted to analyze the gut bacterial change in HCC patients. Fecal counts of *Escherichia coli* (*E. coli.*) increased in 15 cirrhotic HCC patients, when compared to 15 etiology and model for end stage liver disease (MELD) score-matched cirrhosis patients [9]. *E. coli.* and *Enterococcus* increased, while *Bifidobacterium* and *Lactobacillus* significantly decreased in 20 HCC patients vs. 20 normal controls [10].

Recently, metagenomic analysis based on the high-throughput pyrosequencing after amplification of the V3-V4 hypervariable regions of 16S ribosomal ribonucleic acid (rRNA) has become the mainstream method in this field.

A recent Chinese study illustrated the characteristics of the gut microbiome in patients with cirrhotic HBV-HCC, HBV-cirrhosis, and healthy control [11]. Fecal microbial diversity was decreased from healthy controls to cirrhosis, but it was increased from cirrhosis to early HCC with cirrhosis. At the phylum level, *Actinobacteria* was increased in HCC versus cirrhosis. At the genus level, 13 genera including *Gemmiger*, *Parabacteroides*, *Paraprevotella*, and *Clostridium_XVIII* were enriched in early HCC versus cirrhosis. Butyrate-producing genera (*Ruminococcus*, *Oscillibacter*, *Faecalibacterium*, *Clostridium IV*, and *Coprococcus*) were decreased, while lipopolysaccharide (LPS)-producing genera (*Klebsiella* and *Haemophilus*) were increased in early HCC versus controls. Another Chinese study also demonstrated that gut microbial diversity was increased from cirrhosis to HCC [18]. Moreover, the butyrate-producing genera were decreased, while LPS-producing genera were increased in cirrhotic HCC patients. Butyrate, a member of short-chain fatty acids (SCFAs), has beneficial effects in energy metabolism, intestinal homeostasis, and immune responses regulation [19]. Thus, the decrease in butyrate-producing bacteria may contribute to HCC development [11]. LPS can activate toll-like receptor (TLR) 4, produce proinflammatory cytokines (tumor necrosis factor-α [TNF-α], interleukin [IL]-6 and IL-1), and contribute to injury and inflammation-driven tumor promotion [20].

Another Chinese study analyzed the pattern of dysbacteriosis in HCC patients due to different etiologies [12]. Both *Lactobacillus* and *Bifidobacterium* were increased in HCC compared to healthy controls, which was contrary to previous reports. However, the pattern of dysbiosis was different between the HBV-HCC patients and non-HBV non-hepatitis C virus (HCV) HCC (NBNC-HCC) patients. At the phylum level, there was a decrease in *Firmicutes* and an increase in *Proteobacteria* in NBNC-HCC patients, and a decrease of *Proteobacteria* in HBV-HCC patients. NBNC-HCC patients harbored less potential anti-inflammatory bacteria (*Faecalibacterium*, *Ruminococcus*, *Ruminoclostridium*) and more pro-inflammatory bacteria (*Escherichia*-*Shigella*, *Enterococcus*). On the contrary, the HBV-HCC patients harbored more potential anti-inflammatory bacteria (*Faecalibacterium*, *Ruminococcus*, *Ruminoclostridium*). The disparity indicates different biological pathways involved in HCC caused by different etiologies. A recent study showed that *Proteobacteria* were significantly increased in HCC patients vs. healthy controls. Genera of *Proteobacteria*, including *Enterobacter* and *Haemophilus*, were also increased [13]. Given that most pro-inflammatory bacteria come from *Proteobacteria*, this result implied that pro-inflammatory bacteria may be involved in the development of HCC [13].

Apart from Chinese studies, some studies included patients of other nationalities and races. An Italian study identified that *Bacteroides* and *Ruminococcaceae* were increased in the 20 cirrhotic nonalcoholic fatty liver disease (NAFLD)-HCC patients when compared to 20 NAFLD-cirrhosis patients, while *Bifidobacterium*, one species of protective bacteria, was reduced [14]. An Argentine study demonstrated an increase of *Erysipelotrichaceae* family and a decrease of *Leuconostocaceae* family in 25 cirrhotic HCC patients when compared to matched 25 cirrhosis controls. HCC patients possessed an increase of the *bacteriodes*/*prevotella* ratio, decreased genus *Fusobacterium* and *Lachnospiraceae* and increased *Odoribacter* and *Butyricimonas* as well [15]. An Australian study observed an increase of *Enterobacteriaceae* and a decrease of *Eubacteriaceae* at the family level, as well as enriched *Bacteroides caecimuris*, *Veillonella parvula*, *Clostridium bolteae*, and *Ruminococcus gnavus* at species level, in 32 NAFLD-HCC when compared to 28 NAFLD-cirrhosis patients [16]. A Japanese study reported the decrease of gut bacterial diversity was in association with chronic hepatitis C progression [21].

Interestingly, stool sample is not the unique option to be used to study the relationship between microbiota and HCC. Blood sample was used in a recent cross-sectional Korean study to investigate the relationship between circulating microbiota and HCC (both viral and non-viral) for the first time. Microbial diversity in serum was significantly reduced in 79 HCCs when compared to 83 cirrhosis and 201 controls. Moreover, relative abundances of several bacterial taxa were correlated with the presence of HCC. At phylum level, *Firmicutes* was the highest in controls while increased *Proteobacteria* was highest in HCC group. At the genus level, *Pseudomonas* significantly decreased in HCC vs. control. *Staphylococcus*, *Acinetobacter*, *Klebsiella*, and *Trabulsiella* were enriched in HCC [17].

In summary, gut bacteria dysbiosis in HCC patients is widely demonstrated, characterized by lack of protective bacteria and enrichment of harmful ones.

However, inconsistency of study results existed. For example, fecal microbial diversity was decreased from cirrhosis to cirrhotic HCC in some studies [11,18], but reduced in another [17]. Factors such as etiology of liver diseases, demographic characteristics, severity of disease, and sample size may affect the study results, leading to limited credibility and applicability. With regard to the etiology of liver diseases, some studies include merely the HBV-HCC or NAFLD-HCC [11,12,14,16], while others include HCC caused by various causes including HCV, alcohol, non-alcoholic steatohepatitis (NASH), HBV, and others [15,17,18]. Given the possible influence of these etiology on the gut microbiota, the etiology difference may explain, in part, the inconsistency of the dysbiosis pattern of HCC patients between studies [16]. In addition, indices to measure dysbiosis in patients with liver diseases need further exploration. Indices including *Bifidobacterium*/*Enterobacteriaceae* ratio [22], the ratio of autochthonous to non-autochthonous taxa (cirrhosis dysbiosis ratio [CDR]) [23], the Ddys index [13], and *Bifidobacterium*/*Enterococcus* ratio [24] have been reported in previous studies, but head-to-head comparison to figure out the superior is scarce. Further studies are needed to verify the applicability of these indices.

## 3. Mechanism Linking Gut Dysbiosis to HCC

### 3.1. Mechanisms Other Than Bile Acids Dysregulation

More than a decade ago, a mouse model tested the hypothesis that specific intestinal bacteria promote liver cancer in a chemical and viral transgenic mouse model [25]. Underlying mechanisms linking gut dysbiosis to HCC attracted the attention of scientists in the field. So far, leaky gut (a failing gut barrier), bile acids dysregulation, bacterial translocation, endotoxemia and subsequent promotion of liver inflammation, fibrosis, proliferation, and immune suppression have been identified to contribute to the development of HCC in the setting of chronic liver diseases (Figure 1). The concept of the gut–liver axis, bidirectional relationship between the gut and its microbiota, provides the possibility of preventing and treating HCC by targeting gut and its microbiota [16].

The intestinal barrier is formed by multiple layers and tight junctions between enterocytes. Physiologically, the intestinal barrier can prevent liver from exposure to pro-inflammatory MAMPs. Pathologically, MAMPs interact with host PRRs, such as the TLRs, activating downstream pathways and leading to liver inflammation and subsequent liver tumor. Leaky gut, which has been demonstrated in HCC patients commonly by the accumulation of serum LPS, exposes liver to MAMPs [10,26,27], thus, leading to liver inflammation, fibrosis, and tumor.

LPS, the major component of the outer membrane of Gram-negative bacteria, is a common pro-inflammatory MAMP. LPS is associated with HCC development. In a dextran sodium sulfate (DSS)-induced colitis model in mice, either a high-fat diet or a choline-deficient high-fat (CDHF) diet increased systemic LPS levels and promoted HCC formation [28,29]. A low, nontoxic dose of LPS via subcutaneous for 12 weeks osmotic pumps led to a significant increase in inflammatory gene expression, tumor number, and tumor size during diethylamine (DEN), plus carbon tetrachloride (CCl_4_)-induced hepatocarcinogenesis in mice [20]. TLR4 activation in non-bone-marrow-derived resident liver cells by LPS from the intestinal microbiota contributes to injury- and inflammation-driven tumor promotion in a mouse model [20].

Besides the component of Gram-negative bacteria, that of Gram-positive gut microbial component exerts a role in HCC development. The hepatic translocation of obesity-induced lipoteichoic acid (LTA), a Gram-positive gut microbial component, promotes HCC development by creating a tumor-promoting microenvironment. LTA enhances the senescence associated secretory phenotype (SASP) of hepatic stellate cells (HSCs) collaboratively with deoxycholic acid, to upregulate the expression of SASP factors and cyclooxygenase-2 (COX2) through TLR2. COX2-mediated prostaglandin E2 (PGE2) production suppresses the antitumor immunity through a prostaglandin E receptor 4 (PTGER4) receptor, thereby contributing to HCC progression [30].

In addition to systemic inflammatory status, an enhanced intestinal inflammatory status was identified in cirrhotic NASH-HCC patients vs. NAFLD cirrhosis, indicated by increased levels calprotectin in feces. Meanwhile, plasma levels of IL8, IL13, chemokine C-C motif ligand (CCL) 3, CCL4, and CCL5 were higher in the HCC group and associated with an activated status of circulating monocytes, which indicates the correlation between gut microbiota profile and systemic inflammation [14].

Leaky gut also increases hepatic exposure to bacterial metabolites, such as bile acids. Some subsets of bile acids have been known as carcinogens for a long time and have the potential to induce HCC. The role of bile acids–bacterial microbiota in the development of HCC is discussed below.

### 3.2. Bile Acids Dysregulation in HCC

Emerging evidence indicates the association between bile acid–bacterial microbiota crosstalk and the development of HCC. Bile acids, synthesized from cholesterol in the liver, are metabolized by gut bacteria and subsequently sensed by two major sensing receptors, farnesoid X receptor (FXR) and G protein-coupled bile acid receptor 1 (GPBAR1) (transmembrane G-protein-coupled receptor 5, [TGR5]). Bile acids pools comprise a variety of species of amphipathic acidic steroids and have both protective and pathogenic roles in liver diseases. Hydrophilic bile acids, such as ursodeoxycholic acid (UDCA), and its taurine-conjugated form tauroursodeoxycholic acid (TUDCA), show profound cytoprotective properties [5], while excessive production of hydrophobic bile acids is cytotoxic and promotes hepatocyte injury [31].

FXR, a nuclear receptor, activation is involved in regulating antibacterial defense in the small intestine [32], preventing chemically induced intestinal inflammation [33] and modulating liver regeneration [34]. FXR and epidermal growth factor receptor (EGFR) signaling is involved in regulating intestinal cell proliferation by bile acids [35]. In addition, FXR and small heterodimer partner (SHP) can regulate protein N-glycan modifications in the liver [36]. TGR5, a plasma membrane receptor, is expressed in sinusoidal endothelial cells, Kupffer cells, cholangiocytes and activated hepatic stellate cells, modulating microcirculation, inflammation, regeneration, biliary secretion, and gallbladder filling [37].

Multiple genera of the gut microbiota are involved in bile acid metabolism, including *Bacteroides*, *Clostridium*, *Lactobacillus*, *Bifidobacterium*, and *Listeria* in bile acid deconjugation; *Bacteroides*, *Eubacterium*, *Clostridium*, *Escherichia*, *Egghertella*, *Eubacterium*, *Peptostreptococcus*, and *Ruminococcus* in oxidation and epimerization of hydroxyl groups at C3, C7, and C12; *Clostridium* and *Eubacterium* in 7-dehydroxylation; *Bacteroides*, *Eubacterium*, and *Lactobacillus* in esterification; and *Clostridium*, *Fusobacterium*, *Peptococcus*, and *Pesudomonas* in desulfation [38]. Dysbiosis of gut microbiota can affect the bile acids homeostasis theoretically. Indeed, clinical studies and animal models demonstrated the dysbacteriosis of some of these above-mentioned bacteria, dysregulation of bile acids in multiple samples (liver tissue, serum, feces, and urine), and the association between the above two abnormalities [39,40,41,42,43,44,45].

A clinical study showed a decreased abundance of *Bacteroidetes* and *Actinobacteria* on the phylum level in cirrhotic NASH-HCC patients when compared to healthy control. Decreased abundances of *Bacteroides* and *Bifidobacterium* and increase of *Lactobacillus* and *Ruminococcus* on the genus level was shown as well. *Lactobacilli* and *Bacteroides* predominantly express bile salt hydrolase (BSH), a deconjugation enzyme involved in bile acids metabolism [42]. Serum levels of the total bile acids and primary conjugated bile acids (glycocholic acid [GCA], taurocholic acid [TCA], glycochenodeoxycholic acid [GCDA], taurochenodeoxycholic acid [TCDA]) were significantly increased in cirrhotic HCC patients [42].

A streptozotocin and high fat diet (HFD)-induced NASH-HCC mouse model showed the increase of the relative abundance of operational taxonomic units (OTUs) in the fecal *Firmicutes* and *Antinobacteria* and the decrease of *Bacteroides* and *Proteobacteria* at the phylum level; increase of *Clostridium*, *Bacteroides*, and *Desulfovibrio* at the genus level; increase of fecal *Clostridium* spp., *Bacteroides* spp., *Atopobium* spp., and *Desulfovibrio* spp. and decrease of *Paasutterella* spp. and *Akkermansia* spp. at the species level [45]. Meanwhile, this model showed the increase of intrahepatic hydrophobic bile acids including deoxycholic acid (DCA), TCA, taurochenodeoxycholic acid (TCDCA), and taurolithocholic acid (TLCA). More importantly, gut microbiota alteration is correlated with altered bile acids during hepatocarcinogenesis [45].

Some species of bile acids have the potential to induce tumor. For example, DCA was shown to be a carcinogen in mice more than 80 years ago [46]. Although the exact pathways and signals involved in bile acids-induced hepato-carcinogenesis remain to be explored, experiments have demonstrated the relationship of some species of bile acids and the development of HCC. In a mouse model, alterations of gut microbiota were induced by dietary or genetic obesity, which resulted in the increase of DCA enterohepatic circulation. DCA provokes a SASP phenotype in HSCs, which secretes various inflammatory and tumor-promoting factors in the liver and facilitates HCC development [47]. Later, the collaborative role of DCA and LTA, a Gram-positive gut microbial component, was identified in the induction of SASP factors and COX2 expression through TLR2-mediated signaling in senescent HSCs in obesity-associated liver tumors. COX2-mediated PGE2 production facilitate tumor progression by suppressing antitumor immunity [30]. A recent study identified that bile acid was used as a messenger by gut microbiome to regulate hepatic antitumor immunity. Primary bile acids increased C-X-C chemokine ligand 16 (CXCL16) expression, whereas secondary bile acids showed the opposite effect. CXCL16 level on liver sinusoidal endothelial cells (LSEC) can control the accumulation of C-X-C chemokine receptor 6 (CXCR6)^+^ hepatic NKT cells, which have an activated phenotype and inhibit liver tumor growth. Specifically, *Clostridium* species can inhibit the accumulation of natural killer T (NKT) cells by accelerating the conversion of secondary bile acids [48]. GCDA significantly enhance the invasive potential of HCC cells by inducing autophagy through adenosine monophosphate activated protein kinase (AMPK)/mammalian target of rapamycin (mTOR) pathway [39]. In an HFD-induced NASH mouse model, gut microbiota-produced secondary bile acids, such as deoxycholic acid, activated the mTOR pathway in hepatocytes, caused hepatic inflammation and injury, and contributed to carcinogenesis [49]. Expression of tumor suppressor gene CEBPa is downregulated in TCDCA-treated HepG2 cell line [45].

In summary, bile acids and bacterial microbiota are closely related, and the relationship plays an important role in the development of HCC.

## 4. Microbial Dysbiosis in HCC Diagnosis

Both fecal and circulating microbial dysbiosis have the potential value in diagnosis of HCC (Table 2).

Previously, stool sample was used to distinguish HCC from cirrhosis or healthy control. Discrimination of HCC from cirrhosis based on fecal *E. coli.* counts achieved an area under the curve (AUC) of 0.742 (95% confidence interval, 0.564–0.920), with the optimal cutoff on the level of 17.728 (natural logarithm of colony-forming units per 1 g of feces). Sensitivity and specificity rates for the established cutoff value were 66.7% and 73.3%, respectively [9]. In addition, a recent Chinese study reported the optimal 30 OTUs markers achieved an AUC of 0.806 (95% CI, 0.745–0.868) between 75 early HCC and 105 non-HCC samples. Importantly, these microbial markers were validated in patients from other areas of China [11]. Moreover, three biomarkers (*Enterococcus*, *Limnobacter*, and *Phyllobacterium*) were identified for HCC diagnosis with high accuracy (AUC > 0.85). *Enterococcus* could be used as a biomarker between liver cirrhosis (LC) and liver cirrhosis-induced HCC (LC-HCC) and between LC and non-liver cirrhosis-induced HCC (NLC-HCC). *Limnobacter* and *Phyllobacterium* could also be used as biomarkers between LC and NLC-HCC [18]. Thirteen genera were discovered to be associated with the tumor size of HCC [18].

Recently, blood microbial dysbiosis presented the diagnostic value in HCC. A Korean study evaluated the value of serum microbiome-based signatures for the detection of HCC. The model based on 5 OTUs (*Pseudomonas*, *Streptococcus*, *Staphylococcus*, *Bifidobacterium*, and *Trabulsiella*) in blood distinguished HCC from controls, which reached an AUC of 0.879 (sensitivity, 0.729; specificity, 0.850; accuracy, 0.816) and 0.875 (sensitivity, 0.756; specificity, 0.797; accuracy, 0.798) in the model development set and test set, respectively [17].

Clinical and translational studies showed great potential of bile acids alone or together with other biomarkers in the diagnosis and prognosis of HCC [40,41,43]. A panel combined by serum GCA and phenylalanyl-tryptophan (Phe-Trp) was ideal, with an AUC of 0.930, 0.892, and 0.807 in discovery, test, and validation set, respectively, and high sensitivity (range 86.0–92.1%), to distinguish HCC from liver cirrhosis patients [40]. When combined with AFP, a traditional HCC biomarker, this panel provided even better prediction of preclinical HCC before clinical diagnosis [40]. A panel combined by chenodeoxycholic acid (CDCA) and other three metabolite (lysophosphatidylcholine [LPC] 20:5, succinyladenosine and uridine) can distinguish HCC from liver cirrhosis with an AUC score of 0.938, sensitivity of 93.3%, and specificity of 86.7% [41]. The high total bile acid (TBA) level in HCC tissue was associated with more invasive and poor survival in HCC patients [39].

Together, microbiome-based signatures, specific bacterial counts, and bile acids may serve as non-invasive biomarkers for HCC.

## 5. Targeting Microbial Dysbiosis in HCC Treatment and Prevention

The clear role microbial dysbiosis in the development of HCC, offers multiple pathways and targets for HCC treatment and prevention theoretically.

For example, PGE2 and its receptor may be novel therapeutic targets for noncirrhotic NASH-associated HCC [30]. Blocking DCA production or reducing gut bacteria efficiently prevents HCC development in obese mice [47]. Gut sterilization can prevent HCC in a mouse model, suggesting that the intestinal microbiota and TLR4 represent therapeutic targets for HCC prevention in advanced liver disease. TLR antagonists can block the propagation of downstream cytokine release [20,50]. Reduction of HCC development by modulating gut microbiota was showed in animal models [20,26,47]. Antibiotics can be used to eliminate disease-promoting bacteria and decrease release of MAMPs and metabolites from a leaky gut. FXR agonists can modulate various downstream immune-related pathways.

Fecal microbiota transplantation (FMT) is the transfer of stool from a healthy donor into the gastrointestinal tract, aiming to gain a therapeutic benefit by changing or normalizing the recipient’s gut microbiota directly. FMT has been approved for treating recurrent and refractory *Clostridium difficile* infection (CDI) by the United States Food and Drug Administration. In the field of treating liver diseases, FMT can improve neurocognitive function and reduce the readmission of patients with hepatic encephalopathy (HE), despite the small scale of study and absence of long-term follow-up [51]. What is more gratifying is that microbiota originating from donors was found in human recipients one year after FMT [51]. However, clinical study regarding FMT in the treatment and prevention of HCC is still missing.

Probiotics can be used to restore normal microbiota composition, suppress the growth of pathogenic microorganisms, and interact with the mucosal system, which affects the systemic immunity. Administration of a commercial probiotic compound VSL#3 (VSL Pharmaceuticals, Fort Lauderdale, FL, USA) dramatically suppressed penicillin-increased HCC formation in rats [27]. A mouse model demonstrated that the efficacy of a novel probiotic mixture (Prohep) slows down the tumor growth significantly and reduces the tumor size and weight by 40% compared with the control [52]. Notably, Prohep limits tumor growth by reducing angiogenesis, and so forth lead to hypoxia-induced cell death in tumor. This indicates that combining Prohep with drugs of other mechanisms, such as immunotherapy, may play a synergistic therapeutic effect.

Given the BA-bacterial microbiota crosstalk in the development of HCC, restoring bile acids homeostasis by modulating gut microbiota or targeting directly bile acids may be effective strategies on preventing and treating HCC. Treatment with antibiotics dramatically reduced the accumulation of secondary bile acids and significantly suppressed tumor developments in the HFD mouse model [49]. An obese mouse model showed that blocking DCA production or reducing gut bacteria by oral antibiotic caused a marked reduction of HCC development in obese mice [47]. Treatment with antibiotics significantly attenuated liver pathology and suppressed tumor development in a new class NASH-inducing HFD mouse model [49]. In addition, oral administration of cholestyramine, bile acid sequestrant to enhance intestinal excretion of hydrophobic bile acids, significantly prevent HCC in a mouse model [45]. Depleting Gram-positive bacteria by vancomycin treatment can induce hepatic NKT cell accumulation and suppress liver tumor growth in multiple mouse models, while feeding secondary bile acids or colonization of bile acid-metabolizing bacteria can reverse both NKT cell accumulation and inhibition of liver tumor growth in mice [48].

Together, targeting microbial dysbiosis to treat and prevent HCC seems promising. However, there is no clinical data in this regard currently.

## 6. Limitations

The current understanding of the altered gut microbiota in HCC remains incomplete. Most clinical studies are single-centered, with small sample sizes, which undermines the applicability of the results. Multi-center, large-scale, cross-ethical studies are in need. In addition, confounding factors, such as etiology of chronic liver diseases, cirrhotic status, diet, alcohol consumption, antibiotics used to control infections, may influence the characteristic of gut microbiota and complicate the situation. These factors need be considered in the design of future study.

The characteristic of dysbiosis vary among HCC patients due to different causes. Although there are some studies showing the etiology of HCC (HBV, HCV, or alcoholic liver disease [ALD]) was not associated with intestinal microbial dysbiosis in HCC [18], more studies do not agree with it. A study published in 2016 described differences in the gut microbiota between different types of underlying liver disease. Two OTUs, OTU-23 (*Neisseria*) and OTU-36 (*Gemella*), were found discriminative between HBV-LC and primary biliary cirrhosis [53]. Similarly, the bacterial diversity level and composition varied differently between NBNC-HCC and HBV-HCC patients [12], and between HBV-HCC and NAFLD-HCC patients [11,16]. Therefore, for the comprehensive recognition of dysbiosis pattern in HCC, future studies need take into account the etiology of liver diseases.

In addition, a recent study demonstrated that neither the serum levels of total serum bile acids nor primary conjugated bile acids differed between NASH-cirrhosis and cirrhotic NASH-HCC patients, although both of them were significantly increased in cirrhotic NASH-HCC patients compared to the healthy control [42]. This result suggests the necessity and importance to clarify whether the HCC develops on the basis of cirrhosis in future studies.

Fecal microbiota might not reflect the change of bacterial in the upper gastrointestinal tract, while small intestinal bacterial overgrowth (SIBO) may play a role in the progression of chronic liver disease (CLD) patients [54]. More efforts need to be done in this regard.

Importantly, correlation does not imply causation. Though studies regarding association of gut microbiota and HCC are emerging, those regarding causative relationship are very limited [55]. From the perspective of methodology, the mainstream sequencing technology, 16S rRNA sequencing, excludes the rare but important microbial cells such as eukaryotes, which makes the spectrum incomplete and may lead to the missing of important information. Moreover, contamination needs be taken into account when using low-biomass samples, such as blood for subsequent microbiome analysis [56].

Finally, and most importantly, clinical study focusing on prevention and treatment of HCC by microbiota modulation is still missing, despite extensive preclinical evidence. Translation of in vitro or in vivo findings to the human context is always difficult [55].

## 7. Future Directions

An optimized therapeutical strategy may be achieved by harboring protective gut bacteria on the basis of available therapies. Despite missing clinical studies, animal experiments reported the enhanced efficacy by modulating gut bacteria. Optimal responses to cancer therapy require an intact commensal microbiota mediating its effects by modulating myeloid-derived cell functions in the tumor microenvironment [57]. Reconstitution of germ-free mice with fecal material from responding melanoma patients improved tumor control, augmented T cell responses, and achieved greater efficacy of anti-programmed cell death protein 1(PD-1)-based immunotherapy [58]. *Bifidobacterium* spp. enhanced the efficacy of anti-programmed cell death ligand 1(PD-L1) therapy in melanoma [59]. Antitumor effects of anti-cytotoxic T lymphocyte antigen (CTLA)-4 blockade in melanoma depend on distinct *Bacteroides* species [60]. Another study suggested that fecal SCFA concentrations may be used as a potential biomarker to identify patients with solid tumors who could benefit from treatment with PD-1 inhibitors [61]. Given the potential therapeutic effects of immunotherapy for advanced HCC [62], it is valuable to explore which species of gut bacteria can benefit HCC patients from immunotherapy. Indeed, a recent study provided evidence that the gut microbiome could influence the sensitivity of anti-PD-1 immunotherapy in HCC patients [63]. Given the elicited immunosuppressive phenotype in peripheral blood mononuclear cells by bacterial extract from NAFLD-HCC patients, modulation of gut microbiota may be of benefit in overcoming immunotherapy resistance in HCC [16].

Intratumoral bacteria exist in seven cancer types, including breast, lung, ovary, pancreas, melanoma, bone, and brain tumors, and contribute to the response to chemotherapy by expressing enzymes capable of metabolizing the drug in a colon cancer mouse model [64,65]. Whether intratumoral bacteria exist in HCC remained unanswered. Given the anatomy connection of the liver with the gut through the protein vein, the liver seems to more susceptible to gut bacteria. Thus, intratumoral bacteria in HCC is worth exploring.

## 8. Conclusions

In conclusion, gut microbiota contributes to the development of HCC through complicated mechanisms, including leaky gut, bile acids dysregulation, bacterial translocation, endotoxemia and subsequent promotion of liver inflammation, fibrosis, proliferation, and immune suppression. Gut bacterial dysbiosis may be an option for early diagnosis of HCC. Although clinical study on HCC treatment by targeting dysbiosis is missing, animal experiments have shed light on the promising application of gut bacterial modulation on HCC treatment.

## Figures and Tables

**Figure 1 pathogens-10-00782-f001:**
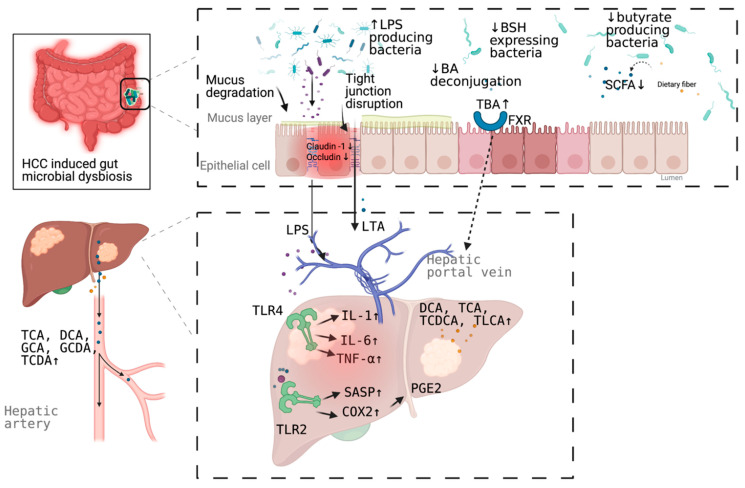
Schematic representation of the mechanism of the promotion and progresssion of HCC by gut microbiota. BA, bile acid; TBA, total bile acid; LPS; Lipopolysaccharides; BSH, bile salt hydrolase; LTA, Lipoteichoic acid; SCFA, short chain fatty acid; DCA, deoxycholic acid; TCA, taurocholic acid; GCA, glycocholic acid; GCDA, glycochenodeoxycholic acid; TCDA, taurochenodeoxycholic acid; TCDCA, taurochenodeoxycholic acid; TLCA, taurolithocholic acid (TLCA); PGE2, prostaglandin E2; COX2, cyclooxygenase-2; SASP, senescence associated secretory phenotype; TLR, toll-like receptor, FXR, farnesoid X receptor. Figure created with BioRender.com (San Francisco, CA, USA).

**Table 1 pathogens-10-00782-t001:** Gut bacteria dysbiosis in HCC patients.

Patients/Control	Increased Microbiota	Decreased Microbiota	Reference
cirrhotic HCC/cirrhosis	*Escherichia coli*.		[9]
HCC/NC	*Escherichia coli.*, *Enterococcus*	*Bifidobacterium*, *Lactobacillus*	[10]
HCC/cirrhosis HCC/cirrhosis HCC/control	*Actinobacteria* *Gemmiger*, *Parabacteroides*, *Paraprevotella*, *Clostridium_XVIII* *Klebsiella* and *Haemophilus*	*Ruminococcus, Oscillibacter, Faecalibacterium*, *Clostridium IV*, and *Coprococcus*	[11]
HCC/NC NBNC-HCC/NC HBV-HCC/NC NBNC-HCC/NC HBV-HCC/NC	*Lactobacillus*, *Bifidobacterium* *Proteobacteria* *Escherichia*-*Shigella*, *Enterococcus* *Faecalibacterium*, *Ruminococcus*, *Ruminoclostridium*	*Firmicutes* *Proteobacteria* *Faecalibacterium*, *Ruminococcus*, *Ruminoclostridium*	[12]
HCC/NC	*Proteobacteria (Enterobacte*, *Haemophilus*)		[13]
NAFLD-HCC/NAFLD-cirrhosis	*Bacteroides*, *Ruminococcaceae*	*Bifidobacterium*	[14]
cirrhotic HCC/cirrhosis	*Erysipelotrichaceae* *Odoribacter*, *Butyricimonas*	*Leuconostocaceae* *Fusobacterium*, *Lachnospiraceae*	[15]
NAFLD-HCC/NAFLD-cirrhosis	*Enterobacteriaceae* *Bacteroides caecimuris*, *Veillonella parvula*, *Clostridium bolteae*, and *Ruminococcus gnavus*	*Eubacteriaceae*	[16]
HCC/NC	*Proteobacteria* *Staphylococcus*, *Acinetobacter*, *Klebsiella*, *Trabulsiella*	*Pseudomonas*	[17]

HBV, hepatitis B virus; HCC, hepatocellular carcinoma; NAFLD, nonalcoholic fatty liver disease; NBNC, non-hepatitis B virus non-hepatitis C virus; NC, normal control.

**Table 2 pathogens-10-00782-t002:** Diagnostic value of microbiota and metabolites in HCC.

Microbiota ^1^	Patients/Control	AUC	95% CI	Sensitivity	Specificity	Reference
*Escherichia coli*	HCC/cirrhosis	0.742	0.564–0.920	66.7%	73.3%	[9]
30 OTUs markers	HCC/non-HCC	0.806	0.745–0.868	-	-	[11]
*Enterococcus*	Cirrhotic HCC/cirrhosis	0.868	-NA	95.8%	69.2%	[18]
*Enterococcus*	Non-cirrhotic HCC/cirrhosis	0.899	NA	100%	78.3%	
*Limnobacter*	Non-cirrhotic HCC/cirrhosis	0.858	NA	62.5%	91.3%	
*Phyllobacterium*	Non-cirrhotic HCC/cirrhosis	0.868	NA	75.0%	91.3%	
5 OTUs markers (serum)	HCC/control	0.879	NA	72.9%	85.0%	[17]
Phe-Trp + GCA (serum)	HCC/cirrhosis	0.807	0.753–0.861	92.1%	52.8%	[40]
Phe-Trp + GCA +AFP (serum)	HCC/cirrhosis	0.826	0.774–0.877	77.9%	76.4%	
CDCA + LPC 20:5 + succinyladenosine + uridine (serum)	HCC/cirrhosis	0.938	-	93.3%	86.7%	[41]

^1^ feces sample is used if not specified. AFP, alpha-fetoprotein; AUC, area under the curve; CDCA, chenodeoxycholic acid; CI, confidence interval; GCA; glycocholate; HCC, hepatocellular carcinoma; LPC, lysophosphatidylcholine; NA, not available; OTU, operational taxonomic unit; Phe-Trp, phenylalanyl-tryptophan. -NA: failed to find out the 95%CI from the paper.

## Data Availability

Not applicable.

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
