# Peer review of "Clinical Aspects of Gut Microbiota in Hepatocellular Carcinoma Management"

_pathogens, 2021, doi:10.3390/pathogens10070782_

Round 1
Reviewer 1 Report
The manuscript entitled "Clinical aspects of gut microbiota in hepatocellular carcinoma management" is an interesting review about a few investigated issue.
Title and abstract are correct, providing enough information to the readers.
English language is correct, requiring a minor review in some parts of the paper.
Attending to the structure, authors provide an easy manner to read the paper as well as to understand.
The different sections provide a general information about the benefits of gut microbiota against HCC as well as a physiological explanation.
Finally, figure is very interesting.
Author Response
Thank you for your support!
Reviewer 2 Report
Within their review, Xu and Zhan et al. give an overview about the current knowledge about implications of intestinal microbiota alterations in the context of hepatocellular carcinoma (HCC).
Overall, this review is nicely structured and gives broad information, with a special focus on bile acid (BA)-associated changes.
The authors should address the following points regarding their manuscript:
Major points
- the article should be proof-read by an English native speaking person
- In its current form, the title “Clinical aspects of gut microbiota in hepatocellular carcinoma management” is misleading since the reader will await more clinically relevant information as it is given in the current form. So, the authors may think about a change of the title that better suits the content of this review
- the authors should integrate data of the work by Inoue et al in this review (https://pubmed.ncbi.nlm.nih.gov/29718124/). In line with this data and as relevant pre-condition for the development of HCC, the authors should give an overview about known associations between liver fibrosis/cirrhosis and the intestinal microbiota dysbiosis
- in addition, the authors should outline the general context of intestinal microbiota alterations in chronic liver diseases within the introduction
- line 157 ff.: relevant data and literature regarding bacterial translocation and the overall concept of the gut-liver axis should be included/integrated and discussed, respectively (e.g. https://pubmed.ncbi.nlm.nih.gov/31622696/ + https://pubmed.ncbi.nlm.nih.gov/23993913/)
- Within the limitations sections, the authors should additionally discuss about the issue of limited data regarding associations/causes/consequences of microbiota alterations and dysbiosis for disease-states (e.g. https://pubmed.ncbi.nlm.nih.gov/29522742/) as well as resulting implications for future. Moreover, the authors should specifically discuss limitations regarding the use of low-biomass samples, such as of blood for subsequent microbiome analysis (especially with regard to metagenomics, contamination etc.; e.g. https://pubmed.ncbi.nlm.nih.gov/30954950/).
- In the “Future directions” section, the authors should include data of Behary et al. about the impact of intestinal microbiota on the peripheral immune response in non-alcoholic fatty liver disease related HCC (https://pubmed.ncbi.nlm.nih.gov/33420074/)
Minor points / spelling / typos etc.
- throughout the manuscript, the authors should adapt correct scientific notation of microorganisms regarding their phylogenetic annotation (e.g. ion https://wwwnc.cdc.gov/eid/page/scientific-nomenclature)
- line 49: Bile acids => bile acids
- line 63 and ff.: is there a specific reason for using the term “dysbateriosis” instead of dysbiosis? If yes, please clarify.
- line 70-72: please rephrase this sentence since it is confusing in its current form
- line 146: “gut leak” should be changed in “leaky gut“ (or bacterial translocation, respectively)
- Figure 1: tight junction proteins (I think they are drawn) should be specifically indicated and named in the figure and the respective figure legend, as they play a major role for gut containment and in bacterial translocation
- line 190 = “species” sounds weird and should be changed (e.g. subsets)
- line 233-238: please rephrase this sentence => meaning and syntax is confusing
Reviewer 3 Report
Comments to the authors
The manuscript entitled “Clinical aspects of gut microbiota in hepatocellular carcinoma management” by Jinghang Xu, et al. reviewed that the impact of the alteration of gut bacterial microbiota in HCC patients and the underling mechanisms of dysbiosis in HCC development, and discussed diagnostic value of gut dysbiosis and therapeutic potential by targeting gut dysbiosis in HCC. Although the review is of interest and important, there are some points to be improved.
Major comments
- At the section of “2. Gut microbiota changes in HCC patients”, the authors summarized some studies illustrating the characteristics of the gut microbiome in patients with liver diseases including cirrhosis or HCC compared healthy controls. The authors should summarize the specific genera in each liver diseases in Table for good understanding.
- At the section of “4. Microbial dysbiosis in HCC diagnosis”, the authors showed that both fecal and circulating microbial dysbiosis had potential value in diagnosis of HCC. The authors should summarize the values of AUC, 95%CI, sensitivity or specificity in each factors in Table for good understanding.
- The authors discussed that it is valuable to explore which species of gut bacteria can benefit HCC patients from immunotherapy, and showed a recent study provided evidence that the gut microbiome could influence the sensitivity of anti-PD-1 immunotherapy in HCC patients [57] at the section of “7. Future directions”. The authors should more reveal and discuss the relationship between immunity and microbial dysbiosis because immune systems have important roles for antitumor effects. Are there any studies illustrating the relationship between the status of hepatic immune cells, such as natural killer cells, cytotoxic T cells, or macrophages, and gut microbiota in various liver diseases?
Minor comments
Nothing to be improved.
Round 2
Reviewer 2 Report
More or less, the authors have adequatly adressed the formerly identified points.
Reviewer 3 Report
The authors politely responded to each revise in new revision.